# *Escherichia coli* Nissle 1917 administered as a dextranomar microsphere biofilm enhances immune responses against human rotavirus in a neonatal malnourished pig model colonized with human infant fecal microbiota

**Husheem Michael[1], Francine C. Paim[1], Ayako Miyazaki[1,2], Stephanie N. Langel[1¤b], David D. Fischer[1¤a], Juliet Chepngeno[1], Steven D. Goodman[3], Gireesh Rajashekara[1], Linda J. Saif[1]\*, Anastasia Nickolaevna Vlasova[1]\***

**1** Food Animal Health Research Program, Department of Veterinary Preventive Medicine, Ohio Agricultural Research and Development Center, The Ohio State University, Wooster, Ohio, United States of America, **2** Division of Viral Disease and Epidemiology, National Institute of Animal Health, National Agriculture and Food Research Organization, Tsukuba, Ibaraki, Japan, **3** Centre for Microbial Pathogenesis, The Nationwide Children's Hospital, Columbus, Ohio, United States of America

¤a Current Address: Present address: Division of Integrated Biomedical Sciences, University of Detroit Mercy School of Dentistry, Detroit, Michigan, United States of America
¤b Current Address: Present address: Duke Human Vaccine Institute, Duke University School of Medicine, Durham, North Carolina, United States of America
\* vlasova.1@osu.edu (ANV); saif.2@osu.edu (LJS)

## Abstract

Human rotavirus (HRV) is a leading cause of diarrhea in children. It causes significant morbidity and mortality, especially in low- and middle-income countries (LMICs), where HRV vaccine efficacy is low. The probiotic *Escherichia coli* Nissle (EcN) 1917 has been widely used in the treatment of enteric diseases in humans. However, repeated doses of EcN are required to achieve maximum beneficial effects. Administration of EcN on a microsphere biofilm could increase probiotic stability and persistence, thus maximizing health benefits without repeated administrations. Our aim was to investigate immune enhancement by the probiotic EcN adhered to a dextranomar microsphere biofilm (EcN biofilm) in a neonatal, malnourished piglet model transplanted with human infant fecal microbiota (HIFM) and infected with rotavirus. To create malnourishment, pigs were fed a reduced amount of bovine milk. Decreased HRV fecal shedding and protection from diarrhea were evident in the EcN biofilm treated piglets compared with EcN suspension and control groups. Moreover, EcN biofilm treatment enhanced natural killer cell activity in blood mononuclear cells (MNCs). Increased frequencies of activated plasmacytoid dendritic cells (pDC) in systemic and intestinal tissues and activated conventional dendritic cells (cDC) in blood and duodenum were also observed in EcN biofilm as compared with EcN suspension treated pigs. Furthermore, EcN biofilm treated pigs had increased frequencies of systemic activated and resting/memory antibody forming B cells and IgA⁺ B cells in the systemic tissues. Similarly, the mean numbers of systemic and intestinal HRV-specific IgA antibody secreting cells (ASCs), as well as HRV-specific IgA antibody titers in serum and small intestinal contents,

**Data Availability Statement:** All relevant data are within the manuscript and its Supporting Information files

**Funding:** This work was supported by the Bill and Melinda Gates Foundation (OPP 1117467), the NIAID, NIH (R01 A1099451), federal and state funds appropriated to the Ohio Agricultural Research and Development Center, The Ohio State University and from the NIH Office of Dietary Supplements (ODS) supplemental grant funds.

**Competing interests:** The authors have declared that no competing interests exist.

were increased in the EcN biofilm treated group. In summary EcN biofilm enhanced innate and B cell immune responses after HRV infection and ameliorated diarrhea following HRV challenge in a malnourished, HIFM pig model.

## Introduction

Human rotavirus (HRV) is a leading cause of diarrhea in children. It causes significant morbidity and mortality, especially in developing countries [1]. Malnutrition is a major contributor of high mortality due to viral gastroenteritis, including HRV, in countries with low socioeconomic status [2–4]. A number of studies have shown that malnutrition triggers immune dysfunction, including altered innate and adaptive immune responses, impairment of epithelial cell barrier function and/or dysfunction of intestinal epithelial cells [5–10].

Probiotics are increasingly used to enhance oral vaccine responses and to treat enteric infections [11] and ulcerative colitis in children [12]. The probiotic *Escherichia coli* Nissle (EcN) 1917 has been widely used in the treatment of ulcerative colitis in humans [13]. EcN lacks virulence factors and possesses unique health-promoting properties [14]. The long term persistence of EcN in humans suggests adaption to a host with an established gut microbiome [15]. Our research group has shown that EcN protected gnotobiotic (Gn) piglets against HRV infection and decreased the severity of diarrhea by modulating innate and adaptive immunity, and protecting the intestinal epithelium [16–18].

Oral administration of probiotics is associated with a number of challenges, such as low pH of gastric acid and bile salts in the stomach, effector functions of the host immune system, and competition with commensal and pathogenic bacteria [19]. These factors adversely influence adherence and persistence of probiotics within the host and thus reduce the beneficial effects [20]. Probiotics must survive in gastric acids to reach the small intestine and colonize the host to confer beneficial effects of preventing or moderating gastrointestinal diseases [21]. Encapsulation of lyophilized probiotics have resulted in enhanced bacterial viability [22, 23]. Navarro and his colleagues (2017) have formulated a new synbiotic formulation that employed porous semi-permeable, biocompatible and biodegradable microspheres (dextranomer microspheres) containing readily diffusible prebiotic cargo [24]. Adherence of the probiotic bacteria to the microsphere has a two-fold effect; it facilitates the more formidable biofilm state of probiotics as well as a creates a directed means to provide a high concentration gradient of prebiotics via diffusion of the microsphere cargo. However, currently there are no strategies for improved EcN probiotic efficacy and stability within the malnourished host.

Previously we have established a deficient HIFM-transplanted neonatal pig model that recapitulates major aspects of malnutrition seen in children in impoverished countries [5, 6]. The purpose of this study was to investigate a novel probiotic delivery method to prolong the persistence of probiotics in the gut and to enhance their beneficial effects. We hypothesized that oral administration of EcN attached to the surface of biocompatible dextranomar microspheres in a biofilm state will protect against harsh conditions of the stomach and improve gut stability, thus enhancing their beneficial effects with a single administration compared with the repetitive administration of probiotics in the suspension form, which results in transient and often inconsistent outcomes. In addition, administration of probiotics in their suspension state has modest impact on the host's microbiome [25]. High doses and repeated administration of probiotics are needed to achieve potential health benefits; however, in impoverished countries this poses challenges due to lack of product availability, the limited health care

system, and resources [26–28]. Whether the use of the biofilm microsphere can overcome this remains to be established.

The multifactorial pathobiology of malnutrition is associated with a vicious cycle of intestinal dysbiosis, epithelial breaches, altered metabolism, impaired immunity, intestinal inflammation, and malabsorption [29, 30]. Malnutrition increases the risk of diarrheal diseases caused by some, but not all, entero-pathogens. Malnutrition can result in impaired immune defenses that compromise gut integrity, and dybiosis that can influence defense against intestinal pathogens in the malnourished host [31].

This in turn limits the ability of probiotics to repair the intestinal epithelium and establish healthy microbiota. These concerns necessitate further research to enhance the stability and persistence of probiotics in malnourished hosts. Probiotics are generally considered safe, however there are some associated risks. These risks are increased if there are chronic medical conditions that weaken the immune system or if there are gut barrier breeches. Possible risks can include: developing an infection, developing resistance to antibiotics, and developing harmful byproducts from the probiotic supplement. Also, in malnourished hosts due to increased intestinal motility, probiotics can be eliminated from the gut faster limiting their beneficial effects [32, 33].

Furthermore, we aimed to investigate whether a single dose of EcN biofilm microspheres enhances immune responses after HRV infection in a malnourished Gn pig model. Previous transplantation of Gn pigs with probiotic bacteria demonstrated upregulated innate and adaptive immune responses following HRV infection [16, 17, 34–37].

In this preliminary study, we report increased innate immune and B cell responses after EcN biofilm treatment that were associated with protection against HRV disease and infection in a neonatal malnourished, HIFM pig model.

## Materials and methods

### Human Infant Fecal Microbiota (HIFM)

The collection and use of HIFM was approved by The Ohio State University Institutional Review Board (IRB). With parental consent, sequential fecal samples were collected from a healthy, two-month-old, exclusively breastfed, vaginally delivered infant. Samples were pooled and diluted to 1:20 (wt/vol) in PBS containing 0.05% (vol/vol) cysteine and 30% glycerol and stored at -80˚C as described previously [5, 6].

### Virus

HRV (VirHRV) Wa strain passaged 25–26 times in Gn piglets was used to orally inoculate piglets at a dose of $1 \times 10^6$ fluorescent focus units (FFU) as described previously [5, 6].

### Preparation of biofilm dextrananomer microspheres

Anhydrous dextranomer microspheres (Sephadex, GE Healthcare Life Sciences, Pittsburgh, PA) were used. Anhydrous microspheres were hydrated in growth medium at 50 mg per ml and autoclaved for 20 min. Autoclaved microspheres were removed from solution on a vacuum filter apparatus and collected via sterile loop into a filter-sterilized 1M solution of sucrose. The microsphere mixture was vortexed and incubated for 24 hours at room temperature (RT). Sugar was removed from solution on a vacuum filter apparatus and collected via sterile loop. The microspheres were then added to EcN [$1 \times 10^9$ colony-forming unit (CFU) per ml], pelleted, washed, and re-suspended in sterile 0.9% saline. EcN was allowed to incubate with the microspheres for 1h at RT to facilitate binding and stored in -80˚C in 30% glycerol. Prior to

use, microspheres were thawed, mixed 1:1 with Natrel and administered orally. For EcN administered as a suspension, $1 \times 10^9$ CFU per ml was pelleted and re-suspended in sterile 0.9% saline in preparation for oral inoculation.

## Animal experiments

The animal experiments were approved by the Institutional Animal Care and Use Committee at The Ohio State University (OSU). Piglets were derived from near-term sows (purchased from OSU specific pathogen-free swine herd) by hysterectomy and maintained in sterile isolators as described previously [38]. For preliminary investigations, neonatal pigs were randomly assigned to three groups: 1) EcN biofilm (n = 3); 2) EcN suspension (n = 4); and 3) control pigs (n = 3). Pigs were fed a deficient diet of 50% ultra-high temperature pasteurized bovine milk diluted with 50% sterile water which contained half of the recommended protein levels (7.5%) that met or exceeded the National Research Council Animal Care Committee's guidelines for calories, fat, protein and carbohydrates in suckling pigs. All pigs were confirmed free from bacterial and fungal contamination prior to HIFM transplantation by aerobic and anaerobic cultures of rectal swabs. Pigs were orally inoculated with 2ml of diluted HIFM stock at 4 days of age (post-HIFM transplantation day, PTD 0). The pigs were colonized orally with EcN biofilm or EcN suspension at PTD 11. Pigs were then challenged with VirHRV [$1 \times 10^6$ FFU, post challenge day (PCD) 0] at PTD 13 and euthanized at PTD 27/PCD 14. Post-VirHRV challenge, rectal swabs were collected daily to assess HRV shedding. Blood, spleen, duodenum, and ileum were collected to isolate mononuclear cells (MNCs) as described previously (31, 35, 36). Jejunum was collected to isolate intestinal epithelial cells (IECs) using modified protocols [18, 39–41]. Serum and small intestinal contents (SIC) were collected to determine the HRV specific and total antibody responses [6, 17, 34, 42, 43].

## Assessment of clinical signs and detection of HRV shedding

Rectal swabs were collected daily post-VirHRV challenge. Fecal consistency was scored as follows; 0, normal; 1, pasty; 2, semi-liquid; and 3, liquid, and pigs with fecal score more than 1 were considered as diarrheic. Rectal swabs were suspended in 2 ml of minimum essential medium (MEM) (Life technologies, Waltham, MA, USA), clarified by centrifugation for 800 × *g* for 10 minutes at 4˚C, and stored at -20˚C until quantification of infectious HRV by a cell culture immunofluorescence (CCIF) assay as previously described [44].

## Isolation of mononuclear cells (MNCs)

Systemic (blood and spleen) and intestinal (duodenum and ileum) tissues were collected to isolate MNCs as described previously [36, 45, 46]. The purified MNCs were re-suspended in E-RPMI 1640. The viability of each MNCs preparation was determined by trypan blue exclusion (≥95%).

## Flow cytometry analysis

Freshly isolated MNCs were stained to assess frequencies of conventional dendritic cells (DCs) (cDCs, SWC3a$^+$CD4$^-$CD11R1$^+$) and plasmacytoid DCs (pDCs, SWC3a$^+$CD4$^+$CD11R1$^-$), MHC II and CD103 marker expression on DCs were used in our experiments. Frequencies of IgA$^+$ B lymphocytes were determined by identifying CD79β and IgA expression in MNCs as reported previously [34]. Similarly, frequencies of memory/resting (CD79β$^+$CD2$^-$CD21$^-$) and activated (CD79β$^+$CD2$^+$CD21$^-$) B cells among systemic and intestinal MNCs were determined as described previously [34]. Appropriate isotype matched control antibodies were included.

Subsequently, 50,000 events were acquired per sample using BD Accuri C6 flow cytometer (BD Biosciences, San Jose, CA, USA). Data were analyzed using C6 flow sampler software.

### NK cytotoxicity assay

Total blood MNCs and K562 cells were used as effector and target cells, respectively. Effector: target cell ratios of 10:1, 5:1, 1:1 and 0.5:1 were used and the assay was done as described previously [47, 48].

### HRV-specific and total antibody responses

The HRV specific and total antibody titers in serum and SIC were detected by enzyme-linked immunosorbent assay (ELISA) as described previously [6, 17, 34, 42, 43]. To determine the intestinal antibody responses, small intestinal contents (SIC) were collected with protease inhibitors in the medium.

### HRV-specific Antibody Secreting Cells (ASCs) responses

HRV and isotype-specific antibody secretion in MNCs isolated from blood, spleen, duodenum and ileum were analyzed by ELISPOT assay as described previously [17, 34, 42, 43].

### Isolation of Intestinal Epithelial Cells (IECs) and extraction of RNA

The IECs were isolated from jejunum (mid gut) using a modified protocol adapted from Paim et al. [18, 49]. The viability and numbers of IECs were determined by the trypan blue exclusion method (70–80%). IECs were stored at −80°C in 500 μl of RNAlater tissue collection buffer (Life technologies, Carlsbad, CA, USA) until further analysis. Total RNA from IECs was extracted using Direct-Zol RNA Miniprep (Zymo Research, Irvine, CA, USA) according to the manufacturer's instructions. The RNA concentrations and purity were measured using Nano-Drop 2000c spectrophotometer (Thermo Scientific, Wilmington, DE, USA).

### Real-time quantitative RT-PCR (qRT-PCR) of CgA, MUC2, PCNA, SOX9 and villin gene mRNA levels in Intestinal Epithelial Cells (IECs)

qRT-PCR was performed using equal amounts of total RNA (75 ng) with Power SYBR Green RNA-to-CT 1 step RT-PCR kit (Applied Biosystems, Foster, CA, USA). The primers for enteroendocrine cells chromogramin A (CgA), goblet cells mucin 2 (MUC2), transient amplifying progenitor cells proliferating cell nuclear antigen (PCNA), intestinal epithelial stem cells transcription factor SRY-box9 (SOX9), enterocytes (villin) and β-actin were based on previously published data [18, 39–41]. Relative gene expression of CgA, MUC2, PCNA, SOX9 and villin were normalized to β-actin and expressed as fold change using the $2^{-\Delta\Delta Ct}$ method [50].

### Statistical analysis

All statistical analyses were performed using GraphPad Prism version 6 (GraphPad software, Inc., La Jolla, CA). $Log_{10}$ transformed isotype ELISA antibody titers that were analyzed using one-way ANOVA followed by Duncan's multiple range test. Data represent the mean numbers of HRV specific antibody secreting cells per $5 \times 10^5$ mononuclear cells and analyzed using non-parametric t-test (Mann-Whitney). HRV shedding and diarrheal analysis were performed using two way ANOVA followed by Bonferroni posttest. $^*P$ values < 0.05, $^{**}P$ values < 0.01, and $^{***}P$ values < 0.001. Error bars indicate the standard error of mean.

## Results

### EcN biofilm treatment reduced fecal HRV shedding and protected malnourished pigs from diarrhea post HRV challenge

Analysis revealed that EcN biofilm treated malnourished pigs had shorter and delayed onset of HRV shedding as compared with the EcN suspension and the control group pigs (Table 1). A significant reduction in fecal virus peak titers shed was observed both in EcN biofilm (GMT = 97.9 FFU/ml) and EcN suspension groups (GMT = 112.0 FFU/ml), as compared with the control pigs (GMT = 439.7 FFU/ml). In addition, EcN biofilm and EcN suspension groups had decreased peak shedding titers at PCD 2 as compared with that of control pigs (S1 Fig). EcN biofilm treatment shortened the mean duration of viral shedding to 2.3 days as compared with 4.0 and 6.0 days in EcN suspension treated and control pigs, respectively (Table 1).

Control pigs developed diarrhea (66.7%) at 3.5 days post HRV challenge, continuing for 1.0 days with mean cumulative fecal score 4.7 (Table 1). Single administration of EcN biofilm microspheres completely protected the pigs from diarrhea (Table 1). However, administration of EcN suspension protected only 50% of the pigs from diarrhea. No significant differences were observed for mean days to diarrheal onset (2.5 days), mean cumulative fecal score (5.5) and the mean duration of diarrhea (1.3 days) when they are compared with those in the control group (Table 1). These findings suggest that administration of EcN biofilm suppressed HRV infection greater than EcN administered in suspension.

### EcN biofilm treatment enhanced natural killer (NK) cell cytotoxicity in blood mononuclear cells (MNCs), increased the frequencies of activated pDCs in systemic and intestinal tissues, and increased activated cDCs in the blood and duodenum

NK cell cytotoxicity in blood MNCs was significantly enhanced in EcN biofilm treatment compared with control pigs (Fig 1A). On the other hand, frequency of apoptotic MNCs were marginally decreased in EcN biofilm (3%) compared with EcN suspension (5%) and control (3.5%) pigs in blood (S2 Fig).

**Table 1. Summary of fecal VirHRV shedding and diarrhea following VirHRV challenge (PCD 1–6).**

| Groups[a] | N | HRV Shedding[b] | | | Diarrhea[c] | | | |
|---|---|---|---|---|---|---|---|---|
| | | Geometric mean of peak titer shed (FFU/ml)[d] | Mean days to onset of HRV shedding | Mean duration of HRV shedding | Diarrhea (%) | Mean cumulative fecal score[e] | Mean days of onset of diarrhea | Mean duration of diarrhea[f] |
| EcN biofilm | 3 | 97.9* | 1.3 | 2.3 | 0 | 4.0* | N/A | N/A |
| EcN suspension | 4 | 112.0** | 1.0 | 4.0 | 50.0 | 5.5** | 2.5 | 1.3 |
| Control | 3 | 439.7*** | 1.0 | 6.0 | 66.7 | 4.7*** | 3.5 | 1.0 |

[a] Gnotobiotic (Gn) pigs were fed deficient diet and were transplanted with human infant fecal microbiota (HIFM) at 4 days of age, post-HIFM transplantation day (PTD) 0. Pigs were colonized with a single dose of EcN biofilm and EcN suspension at PTD 11, subsequently challenged with virulent human rotavirus (VirHRV) at PTD 13 and euthanized at PTD27/post challenge day (PCD) 14.

[b] Determined by cell culture immunofluorescence (CCIF) assay and expressed as FFU/ml.

[c] Pigs with fecal score > 1 were considered as diarrheic. Fecal consistency was scored as follows: 0, normal; 1, pasty; 2, semi-liquid; and 3, liquid.

[d] Samples negative for HRV detection (< 25) were assigned a titer of 12.5 for statistical analysis. Means in the same column with different asterisks differ significantly (determined by two-way ANOVA followed by Bonferroni posttest, $p < 0.05$).

[e] Mean of total of fecal scores from PCD 1–6.

[f] Mean of total days with fecal score > 1.

N/A = data not available.

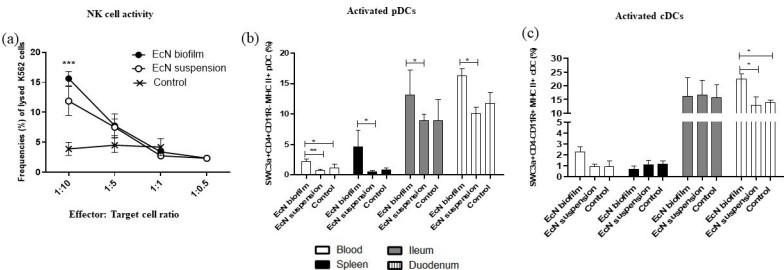

**Fig 1. EcN biofilm enhanced NK cell activity in blood mononuclear cells (MNCs) and significantly increased the frequencies of activated pDCs in systemic and intestinal tissues and increased activated cDCs in blood and duodenum (significantly). (a)** Blood MNCs and carboxyfluorescein diacetate succinimidyl ester (CFSE) stained K562 tumor cells were used as effector and target cells, respectively, and co-cultured at set ratios to assess the NK cytotoxic function, (EcN biofilm vs control group). The effector: target cell co-cultures were stained with 7-Aminoactinomycin D (7AAD) after 12 hours of incubation at 37˚C, and the frequencies of CFSE-7AAD double positive cells (lysed K562 target cells) were assessed by flow cytometry. Mean frequencies of activated **(b)** pDCs and **(c)** cDCs in systemic and intestinal tissues. Data represent means ± SEM. Significant differences (*p < 0.05, **p < 0.01, ***p < 0.001) are indicated. Gnotobiotic pigs were transplanted with human infant fecal microbiota (HIFM) at 4 days of age, post-HIFM transplantation day (PTD) 0. Pigs were fed a deficient diet. Probiotic was given to the respective groups at PTD 11, followed by challenge with virulent human rotavirus (HRV) on PTD 13/post-challenge day (PCD) 0 and pigs were euthanized on PTD 27/PCD 14.

EcN biofilm treatment significantly increased the frequencies of activated pDC in systemic and intestinal tissues as compared with EcN suspension and the control pigs (Fig 1B). Moreover, EcN biofilm treatment significantly increased the frequencies of activated cDC in duodenum while numerically in blood (Fig 1C). There were no differences observed in other tissues. CD103+ cDC were increased (numerically) in spleen and intestinal tissues in EcN biofilm treated group as compared with EcN suspension and control pigs (S3 Fig). There were no differences observed in blood.

## EcN biofilm treatment significantly increased the frequencies of activated antibody secreting B cells in systemic tissues, resting antibody forming B cells in blood, and IgA+ B cells in spleen

EcN biofilm treated malnourished pigs had significantly increased frequencies of activated antibody forming B cells in systemic tissues as compared with EcN suspension or the control pigs (S4A and S4B Fig). The frequency of IgA+ B cells in the spleen (significantly, S4C Fig) and blood (numerically, S4D Fig) increased in EcN biofilm treatment compared with EcN suspension and control pigs. Moreover, the frequency of resting/memory antibody forming B cells was significantly increased in blood in EcN biofilm compared with EcN suspension treated pigs (S4E Fig). These findings suggest that EcN biofilm treatment enhanced B cell immune response in systemic tissues, although no significant trends were observed in intestinal tissues.

## EcN biofilm treatment increased the number of HRV-specific Antibody Secreting Cells (ASCs) in systemic and intestinal tissues, and increased HRV-specific IgA antibody titers in serum and Small Intestinal Contents (SIC)

Coinciding with decreased HRV shedding and protection from diarrhea, the mean numbers of HRV-specific IgA ASCs were increased in systemic and intestinal tissues of EcN biofilm treatment compared with EcN suspension and control group pigs (Fig 2A and 2B). A similar trend was observed with HRV-specific IgG ASCs (S5 Fig). HRV-specific IgM ASC numbers were

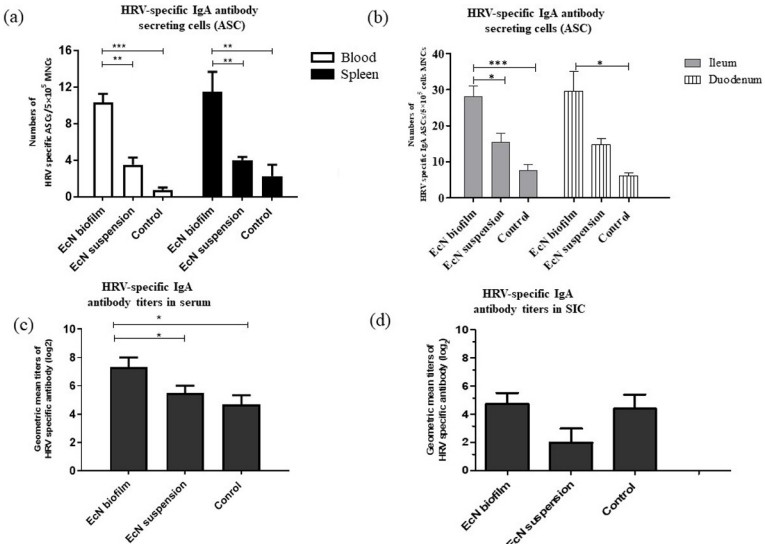

**Fig 2. EcN biofilm significantly increased HRV-specific IgA Antibody Secreting Cells (ASCs) in systemic and intestinal tissues and increased HRV-specific IgA antibody titers in serum and Small Intestinal Contents (SIC).** **(a)** HRV-specific IgA ASCs in systemic cells; **(b)** HRV-specific IgA ASCs in intestinal cells; **(c)** HRV-specific IgA antibody titers in serum and **(d)** SIC. No significant differences were observed in intestinal tissues. Data represent means ± SEM. Significant differences (*p < 0.05, **p < 0.01, ***p < 0.001) are indicated. Gnotobiotic pigs were transplanted with human infant fecal microbiota (HIFM) at 4 days of age, post-HIFM transplantation day (PTD) 0. Pigs were fed a deficient diet. Probiotic was given to respective groups at PTD 11, followed by challenge with virulent human rotavirus (HRV) on PTD 13/post-challenge day (PCD) 0 and pigs were euthanized on PTD 27/PCD 14.

below the detection limit in systemic and intestinal tissues. HRV-specific IgA antibody titers were increased in serum (significantly) and SIC (numerically) of EcN biofilm treated pigs compared with EcN suspension and control group pigs, coinciding with increased HRV-specific IgA ASCs (Fig 2C and 2D). Similar trends were observed with HRV-specific IgG antibody titers in serum (S6 Fig). In addition, total IgA concentration was increased (numerically) in serum samples of EcN biofilm treated pigs compared with EcN suspension or control group pigs (S7 Fig). No significant trends were observed in total and HRV-specific IgG in SIC (S8 Fig). These results indicate that EcN biofilm treatment enhanced B cell formation and clonal expansion of antibody producing cells in malnourished, HIFM transplanted pigs infected with HRV.

## EcN biofilm treatment significantly upregulated the expression of CgA and SOX9 mRNA levels in jejunal epithelial cells

Gene expression levels of CgA, SOX9, villin, MUC2, and PCNA were assessed from jejunal epithelial cells. The relative mRNA levels of CgA, SOX9, and villin genes were increased in jejunal epithelial cells of EcN biofilm compared with EcN suspension and control treated malnourished pigs (Fig 3A–3C). This coincided with the decreased severity of HRV shedding and diarrhea. There were no differences in gene expression levels for MUC2 and PCNA in jejunal epithelial cells of EcN biofilm and EcN suspension treated pigs (S9 Fig).

## Discussion

Using a malnourished and HIFM transplanted pig model, we showed that compared with EcN administered as suspension, EcN administered as a biofilm on dextranomer microspheres enhanced multiple aspects of the immune response. EcN biofilm treated pigs had significantly reduced titers of virus shedding and diarrhea following VirHRV challenge

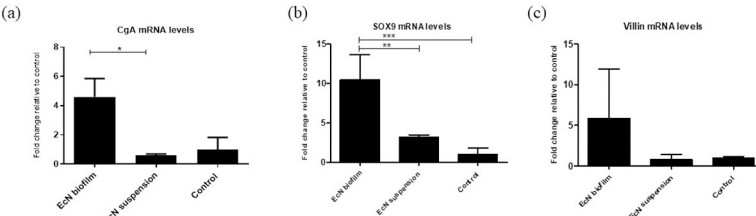

**Fig 3. EcN biofilm upregulated the expression of various cell components in jejunal epithelial cells. (a)** Relative mRNA levels of enteroendocrine cells chromogramin A (CgA), **(b)** intestinal epithelial stem cells (SOX9), and **(c)** enterocytes (villin) in EcN biofilm, EcN suspension groups measured by real-time quantitative RT-PCR (RT-PCR), normalized to β-actin gene. Graphs represent means ± SEM. Significant difference (*p < 0.05, **p < 0.01, relative to control) are indicated. Gnotobiotic pigs were transplanted with human infant fecal microbiota (HIFM) at 4 days of age, post-HIFM transplantation day (PTD) 0. Pigs were fed a deficient diet. Probiotic was given to respective groups at PTD 11, followed by challenge with virulent human rotavirus (VirHRV) on PTD 13/post-challenge day (PCD) 0 and pigs were euthanized on PTD 27/PCD 14.

compared with EcN suspension treated and control pigs. The presence of HRV-specific IgA antibodies in pigs is strongly correlated with protection from HRV infection [46, 51, 52]. Moreover, our study demonstrated for the first time that EcN biofilm treatment enhanced HRV specific-IgA and IgG ASCs in circulation and gut, enhanced HRV-specific IgA and IgG antibody titers in serum and HRV-specific IgA antibody titers in SIC, which collectively coincided with reduced diarrhea and virus shedding. Total IgA concentration was marginally increased in serum of EcN biofilm treated malnourished pigs (data not shown). Although not examined in this study, EcN biofilm treatment might have increased colonization in the gut, inhibiting competition by other members of the gut microbiota [53, 54]. It is possible that the observed effects of EcN biofilm treatment on systemic IgA responses could be mediated by direct modulation of host immune responses, suggesting that biofilm microspheres maybe more stable and persistent compared to probiotics in suspension in the host's gastrointestinal system.

Innate immune responses are critical as a first line of defense, limiting RV replication and disease severity in the host [16, 55]. EcN biofilm treatment enhanced innate immune responses. For example, blood NK cell cytotoxicity was higher in EcN biofilm treatment compared to EcN suspension treated and control groups. This suggests that EcN as a biofilm promoted innate immune responses, improving protection against HRV infection *in vivo*. Also the frequency of apoptotic blood MNCs was slightly reduced in EcN biofilm treated pigs compared with EcN suspension treatment and control pigs (data not shown).

DCs play a key role in probiotic bacteria stimulation of the innate immune system [56, 57] and pDCs were shown to contribute to RV clearance in a murine model [58]. Moreover, DC MHC II expression is a marker for maturation [59]. In our study, higher frequencies of activated pDCs in systemic and intestinal tissues and activated cDCs in the blood and duodenum were observed in EcN biofilm treated pigs compared with EcN suspension treated piglets. These results suggest that the biofilm provided stability to the probiotic and thus enhanced maturation of systemic and intestinal activated DC, promoting pDC development and increased IgA antibody responses in probiotic biofilm treated piglets compared with probiotic suspension treated pigs [60, 61]. Enhancing the protective effects of pDCs via an EcN biofilm may be critical for protection against enteric pathogens [16]. Expression of CD103 ($\alpha_E\beta_7$ integrin) has been demonstrated to influence cellular intraepithelial morphogenesis and motility [62], which are critical for the proper communication among pathogen, DCs, and T and B lymphocytes. We observed that EcN biofilm treatment increased CD103 expression by DCs and this could have further enhanced innate immune responses

against HRV and reduced HRV infection. Consequently, enhancement of signaling between DCs and T/B lymphocytes could have contributed to improved antigen presentation to the lymphocytes resulting in increased HRV-specific IgA ASCs, IgA antibody titers, and increased NK cell activity in EcN biofilm treated pigs.

The increased frequencies of activated and resting/memory B cells were enhanced in EcN biofilm treated pigs that coincided with increased frequencies of pDCs in the intestine. These results are similar to our previous studies where EcN protected against HRV infection [34, 37]. The frequency of IgA+ B cells were increased in EcN biofilm treated pigs in systemic tissues, suggesting that EcN as a biofilm may potentiate systemic IgA responses. These responses and the increased HRV-specific IgA antibody responses in serum and SIC coincided with reduced HRV diarrhea and shedding.

An upregulation of the enteroendocrine CgA gene in EcN biofilm treated piglets could be reflective of greater protection of the epithelial intestinal barrier. Other studies have shown that enteroendocrine cells that produce hormones promoting repair of intestinal epithelium are activated after treatment with probiotics [63, 64]. In our investigations, we observed an upregulation of stem cell specific-gene SOX9 in the EcN biofilm treated pigs greater than in EcN suspension treated pigs. SOX9 plays an important role in the proliferative capacity of stem cells to replenish different lineages of IECs [65]. Moreover, we demonstrated that EcN biofilm treatment increased mRNA levels of the enterocyte-specific gene villin. It is likely that biofilm microspheres supported a greater number of villin cells and epithelial cells and probiotic adherence. This likely modulated the effects of HRV infection by increasing villin gene expression of enterocytes, repairing/restoring functional enterocytes and increasing barrier and absorptive functions during HRV-induced diarrhea.

Our results suggest that using a microsphere biofilm as a novel delivery system for EcN compared to EcN as a suspension may have increased survival of the probiotics at low pH in the stomach and supported increased adherence to intestinal epithelial cells [24], thereby promoting probiotic longevity, survival, and persistence in the malnourished host. Additionally, the EcN biofilm enhanced innate and B cell immune responses in the HRV infected HIFM neonatal pigs. Our results support previous work demonstrating protection against experimental necrotizing enterocolitis in a rat model after treatment with *Lactobacillus reuteri* adhered to dextranomer microspheres [66]. Recently, Shelby et al. 2020 and colleagues have demonstrated that a single dose of *Lactobacillus reuteri* in its biofilm state reduces the severity and incidence of experimental *C. difficile* infection and necrotizing enterocolitis when administered as both prophylactic and treatment therapy [67, 68]. Moreover, Navarro and colleagues demonstrated that probiotic bacterium *L. reuteri* delivered in association with dextranomar microspheres adhered in greater numbers, conferred resistance to clearance, transported nutrients that promote bacterial growth, promoted the production of the antimicrobial reuterin or histamine, resisted acid-mediated killing, and better supported adherence to intestinal epithelial cells, thereby promoting persistence in the gut [24]. Thus, we this agreed with our hypothesis that EcN adhered to dextranomer microspheres acted similarly during HRV infection in the neonatal malnourished HIFM pig model. In the future, we have plan to increase to number of piglets and study different age groups to further investigate the biofilm impact.

Thus, our results suggest that low cost, stable, and efficient dietary supplementation of EcN coupled with a dextranomer microsphere biofilm can protect against HRV infection in a physiologically relevant malnourished HIFM pig model. Similar studies are warranted in children to moderate the symptoms of other gastrointestinal infections and disorders including gastritis and chronic inflammatory bowel disease.

## Supporting information

**S1 Fig. HRV shedding post-VirHRV challenge in EcN biofilm, EcN suspension, and control pigs.**
(PPTX)

**S2 Fig. Frequencies of apoptotic MNCs (PI$^-$/Annexin-APC$^-$) among total MNCs isolated from blood.**
(PPTX)

**S3 Fig. EcN biofilm altered the frequencies of SWC3a$^+$CD4$^-$CD103$^+$ conventional dendritic cells in blood and intestinal tissues.**
(PPTX)

**S4 Fig. EcN biofilm significantly increased the frequencies of activated B cells in systemic tissues and IgA$^+$ B cells in spleen.**
(PPTX)

**S5 Fig. EcN biofilm significantly increased HRV-specific IgG antibody secreting cells (ASCs) in systemic.**
(PPTX)

**S6 Fig. EcN biofilm increased geometric mean titer HRV specific IgG in serum.**
(PPTX)

**S7 Fig. EcN biofilm increased total Ig concentration in serum.**
(PPTX)

**S8 Fig.** Total (a) and HRV-specific IgG (b) in EcN biofilm, EcN suspension, and control pigs in small intestinal contents (SIC).
(PPTX)

**S9 Fig.** Relative mRNA levels of MUC2 (a) and PCNA (b) in EcN biofilm, EcN suspension, and control group measured by real-time quantitative RT-PCR (RT-PCR), normalized to β-actin gene.
(PPTX)

## Acknowledgments

We thank Marcia Lee and Rosario Candelero-Rueda for their technical assistance and Dr. Juliette Hanson, Ronna Wood, Jeffery Ogg, Megan Strother and Sara Tallmadge for animal care assistance.

## Author Contributions

**Conceptualization:** Anastasia Nickolaevna Vlasova.

**Data curation:** Husheem Michael, Francine C. Paim, Ayako Miyazaki, Stephanie N. Langel, David D. Fischer, Juliet Chepngeno.

**Formal analysis:** Husheem Michael, Anastasia Nickolaevna Vlasova.

**Funding acquisition:** Gireesh Rajashekara, Linda J. Saif.

**Investigation:** Gireesh Rajashekara, Linda J. Saif, Anastasia Nickolaevna Vlasova.

**Methodology:** Gireesh Rajashekara, Linda J. Saif, Anastasia Nickolaevna Vlasova.

**Project administration:** Gireesh Rajashekara, Linda J. Saif, Anastasia Nickolaevna Vlasova.

**Supervision:** Linda J. Saif.

**Validation:** Steven D. Goodman, Gireesh Rajashekara, Linda J. Saif, Anastasia Nickolaevna Vlasova.

**Writing – original draft:** Husheem Michael.

**Writing – review & editing:** Francine C. Paim, Ayako Miyazaki, Stephanie N. Langel, David D. Fischer, Juliet Chepngeno, Steven D. Goodman, Gireesh Rajashekara, Linda J. Saif, Anastasia Nickolaevna Vlasova.

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
