## [Decision Letter · Decision Letter 0]

21 Dec 2020

PONE-D-20-32708

Escherichia coli Nissle 1917 administered as a dextranomar microsphere biofilm enhances immune responses against human rotavirus in a neonatal malnourished pig model colonized with human infant fecal microbiota

PLOS ONE

Dear Dr. Vlasova,

Thank you for submitting your manuscript to PLOS ONE. After careful consideration, we feel that it has merit but does not fully meet PLOS ONE’s publication criteria as it currently stands. Therefore, we invite you to submit a revised version of the manuscript that addresses the points raised during the review process.

We look forward to receiving your revised manuscript.

Kind regards,

Nicholas J Mantis

Academic Editor

PLOS ONE

Journal Requirements:

"This work was supported by the Bill and Melinda Gates Foundation (OPP 1117467), the NIAID,

 NIH (R01 A1099451), federal and state funds appropriated to the Ohio Agricultural Research and

Development Center, The Ohio State University and from the NIH Office of Dietary Supplements

(ODS) supplemental grant funds."

"NO: The funders had no role in study design, data collection and analysis, decision to publish, or preparation of the manuscript."

Reviewers' comments:

Reviewer's Responses to Questions

**Comments to the Author**

1. Is the manuscript technically sound, and do the data support the conclusions?

Reviewer #1: Yes

2. Has the statistical analysis been performed appropriately and rigorously? 

Reviewer #1: Yes

3. Have the authors made all data underlying the findings in their manuscript fully available?

Reviewer #1: Yes

4. Is the manuscript presented in an intelligible fashion and written in standard English?

Reviewer #1: Yes

5. Review Comments to the Author

Reviewer #1: This manuscript study the effect of EcN biofilm in Gp pigs with induced nutritional deficiencies and modified gut microbiota. The results are interesting and promising. The manuscript is clear, well described and easy to follow. I just have some minor observations:

Lines 84-87: Please consider to include in this section some information about microparticles and materials studied or proved useful to improve probiotics viability (as explanation of the hypothesis).

Lines 88-100: Besides gut stability, Why a single administration (EcN-biofilm) would be considered enough?

Lines 101-105: In some cases, with the use of probiotics is recommended the use of prebiotics and supplements (to restore gut balance). Explain better the impact of malnutrition in diarrheic diseases and the limitations of the probiotics’ beneficial effects.

Lines 123-125: Why did you decided to use HRV strain Wa instead OSU? Is just because the gut microbiota modification?

Lines 321-323: Although EcN biofilm showed better results, EcN suspention also had good response controlling viral shedding respect to control (and in other parameters). Is the cost-effect good or better in the use of EcN biofilm (production)?

Line 393: more than speculate use other expression… change it by -this would explain- or -this agreed with our hypothesis-

References: Check the references format; in reference 13, the year was missed. Most of the references are old (20% are from 2015-2020)

6. PLOS authors have the option to publish the peer review history of their article (what does this mean?). If published, this will include your full peer review and any attached files.

Reviewer #1: No

---

## [Author Response · Author response to Decision Letter 0]

11 Jan 2021

12.29.2020

Manuscript ID: PONE-D-20-32708 

Dear Editor, 

We sincerely thank you for the opportunity to re-submit our revised manuscript and the reviewer for the constructive and insightful criticism and the overall encouraging comments. Please find below our point-by-point answers to the reviewer’s queries and modifications/insertions are highlighted in the manuscript.

ED: Please ensure that your manuscript meets PLOS ONE's style requirements, including those for file naming. The PLOS ONE style templates can be found at https://journals.plos.org/plosone/s/file?id=wjVg/PLOSOne_formatting_sample_main_body.pdf and https://journals.plos.org/plosone/s/file?id=ba62/PLOSOne_formatting_sample_title_authors_affiliations.pdf

AU: We have reviewed the PLOS ONE guidelines and believe our manuscript meets them.

ED: Please provide additional details regarding participant consent. In the ethics statement in the Methods and online submission information, please ensure that you have specified what type you obtained (for instance, written or verbal, and if verbal, how it was documented and witnessed). If your study included minors, state whether you obtained consent from parents or guardians. If the need for consent was waived by the ethics committee, please include this information.

AU: We have obtained a written consent from parent that is being securely stored. 

ED: Thank you for stating the following in the Acknowledgments Section of your manuscript:

"This work was supported by the Bill and Melinda Gates Foundation (OPP 1117467), the NIAID,

 NIH (R01 A1099451), federal and state funds appropriated to the Ohio Agricultural Research and

Development Center, The Ohio State University and from the NIH Office of Dietary Supplements

(ODS) supplemental grant funds."

"NO: The funders had no role in study design, data collection and analysis, decision to publish, or preparation of the manuscript."

AU: Funding information is removed from the manuscript. During the original manuscript submission, we have entered the funding sources; however, we failed to find where Funding statement could be entered. Please enter the following statement into the online submission system: 

This work was supported by the Bill and Melinda Gates Foundation (OPP 1117467), the NIAID, NIH (R01 A1099451), federal and state funds appropriated to the Ohio Agricultural Research and Development Center, The Ohio State University and from the NIH Office of Dietary Supplements (ODS) supplemental grant funds.

ED: We note that you have included the phrase “data not shown” in your manuscript. Unfortunately, this does not meet our data sharing requirements. PLOS does not permit references to inaccessible data. We require that authors provide all relevant data within the paper, Supporting Information files, or in an acceptable, public repository. Please add a citation to support this phrase or upload the data that corresponds with these findings to a stable repository (such as Figshare or Dryad) and provide and URLs, DOIs, or accession numbers that may be used to access these data. Or, if the data are not a core part of the research being presented in your study, we ask that you remove the phrase that refers to these data.3. 

AU: To comply with the PLOS policy we have now submitted all the data previously labeled as ‘not shown’ as supplementary figures and numbered accordingly. 

 Reviewer # 1 Comments to the Author 

RE: Lines 84-87: Please consider to include in this section some information about microparticles and materials studied or proved useful to improve probiotics viability (as explanation of the hypothesis).

AU: We have now added the following information: Navarro and his colleagues (2017) have formulated a new synbiotic formulation that employed porous semi-permeable, biocompatible and biodegradable microspheres (dextranomer microspheres) containing readily diffusible prebiotic cargo. Adherence of the probiotic bacteria to the microsphere has a two-fold effect; it facilitates the more formidable biofilm state of probiotics as well as a creates a directed means to provide a high concentration gradient of prebiotics via diffusion of the microsphere cargo.

 Now lines 85-91.

RE: Lines 88-100: Besides gut stability, Why a single administration (EcN-biofilm) would be considered enough?

AU: Biofilms are the native stable state of bacteria on surfaces and provide a more protected state of the resident bacteria. They also allow bacteria to evade innate immune defenses and persist in the gut longer (Tam, Uyen et al., 2006; Wang, Cao et al. 2020). These biofilm features allow maximization of the beneficial effects of probiotics following single administration.

RE: Lines 101-105: In some cases, with the use of probiotics is recommended the use of prebiotics and supplements (to restore gut balance). Explain better the impact of malnutrition in diarrheic diseases and the limitations of the probiotics’ beneficial effects. 

AU: Malnutrition increases the risk of diarrheal diseases caused by some, but not all, entero-pathogens. Malnutrition can result in impaired immune defenses that compromise gut integrity, and dybiosis that can influence defense against intestinal pathogens in the malnourished host. 

Now lines 108-111.

Probiotics are generally considered safe, however there are some associated risks. These risks are increased if there are chronic medical conditions that weaken the immune system or if there are gut barrier breeches. Possible risks can include: developing an infection, developing resistance to antibiotics, and developing harmful byproducts from the probiotic supplement. Also, in malnourished hosts due to increased intestinal motility, probiotics can be eliminated from the gut faster limiting their beneficial effects. 

Now lines 114-119. 

RE: Lines 321-323: Although EcN biofilm showed better results, EcN suspension also had good response controlling viral shedding respect to control (and in other parameters). Is the cost-effect good or better in the use of EcN biofilm (production)?

AU: Results suggest that low-cost, stable, and efficient dietary supplementation of EcN coupled with a dextranomer microsphere biofilm can protect against HRV infection and can moderate the symptoms of other gastrointestinal infections and disorders including gastritis and chronic inflammatory bowel disease. Probiotics in the planktonic state need to be taken daily, whereas this biofilm formulation only needs to be taken once which can reduce the cost, despite the extra expense for microsphere formulation.

RE: Lines 123-125: Why did you decided to use HRV strain Wa instead OSU? Is just because the gut microbiota modification?

AU: This is because we were modeling childhood malnutrition-rotavirus infection-probiotic intervention in gnotobiotic pigs transplanted with human infant microbiota and infected with human rotavirus strain (Wa). 

RE: Line 393: more than speculate use other expression… change it by -this would explain- or -this agreed with our hypothesis-

AU: Word ‘speculate’ is replaced with ‘this agreed with our hypothesis’ and now line 395.

RE: References: Check the references format; in reference 13, the year was missed. Most of the references are old (20% are from 2015-2020)

AU: Year is added into the reference, now line 503. Older references including the primary references can be highly relevant—our focus was to cite the most relevant references!

---

## [Editor Report · Decision Letter 1]

15 Jan 2021

Escherichia coli Nissle 1917 administered as a dextranomar microsphere biofilm enhances immune responses against human rotavirus in a neonatal malnourished pig model colonized with human infant fecal microbiota

PONE-D-20-32708R1

Dear Dr. Vlasova,

We’re pleased to inform you that your manuscript has been judged scientifically suitable for publication and will be formally accepted for publication once it meets all outstanding technical requirements.

Kind regards,

Nicholas J Mantis

Academic Editor

PLOS ONE
---

## [Editor Report · Acceptance letter]

25 Jan 2021

PONE-D-20-32708R1 

*Escherichia coli* Nissle 1917 administered as a dextranomar microsphere biofilm enhances immune responses against human rotavirus in a neonatal malnourished pig model colonized with human infant fecal microbiota 

Dear Dr. Vlasova:

I'm pleased to inform you that your manuscript has been deemed suitable for publication in PLOS ONE. Congratulations! Your manuscript is now with our production department. 

Kind regards, 

on behalf of

Dr. Nicholas J Mantis 

Academic Editor

PLOS ONE